# Learning Global Additive Explanations for Neural Nets Using Model Distillation

## Abstract

Interpretability has largely focused on local explanations, i.e. explaining why a model made a particular prediction for a sample. These explanations are appealing due to their simplicity and local fidelity. However, they do not provide information about the general behavior of the model. We propose to leverage model distillation to learn global additive explanations that describe the relationship between input features and model predictions. These global explanations take the form of feature shapes, which are more expressive than feature attributions. Through careful experimentation, we show qualitatively and quantitatively that global additive explanations are able to describe model behavior and yield insights about models such as neural nets. A visualization of our approach applied to a neural net as it is trained is available at `https://youtu.be/ATNcgurNHhc`.

## 1 Introduction

Recent research in interpretability has focused on developing *local* explanations: given an existing model and a sample, explain why the model made a particular prediction for that sample (Ribeiro et al., 2016). The accuracy and quality of these explanations have rapidly improved, and they are becoming important tools to understand model decisions for individual samples. However, the human cost of examining multiple local explanations can be prohibitive with today's large data sets, and it is unclear whether multiple local explanations can be aggregated without contradicting each other (Ribeiro et al., 2018; Alvarez-Melis & Jaakkola, 2018).

In this paper, we are interested in *global* explanations that describe the overall behavior of a model. While usually not as accurate as local explanations on individual samples, global explanations provide a different, complementary view of the model. They allow us to clearly visualize trends in feature space, which is useful for key tasks such as understanding which features are important, detecting unexpected patterns in the training data and debugging errors learned by the model.

We propose to use model distillation techniques (Bucilua et al., 2006; Hinton et al., 2015) to learn global additive explanations of the form

$$\hat{F}(\mathbf{x}) = h_0 + \sum_i h_i(x_i) + \sum_{i \neq j} h_{ij}(x_i, x_j) + \sum_{i \neq j} \sum_{j \neq k} h_{ijk}(x_i, x_j, x_k) + \cdots \tag{1}$$

to approximate the prediction function of the model $F(\mathbf{x})$. Figure 1 illustrates our approach. The output of our approach is a set of $p$ feature shapes $\{h_i\}_1^p$ that can be composed to form an explanation model that can be quantitatively evaluated. Through controlled experiments, we empirically validate that feature shapes provide accurate and interesting insights into the behavior of complex models. In this paper, we focus on interpreting $F$ from fully-connected neural nets trained on tabular data.

Our goal is not to replace local explanations nor to explain how the model functions internally. What we claim is that we can complement local explanations with global additive explanations that clearly illustrate the relationship between input features and model predictions. Our contributions are:

- We propose to *learn* global additive explanations for complex, non-linear models such as neural nets.
- We leverage powerful generalized additive models in a model distillation setting to learn feature shapes that are more expressive than feature attributions

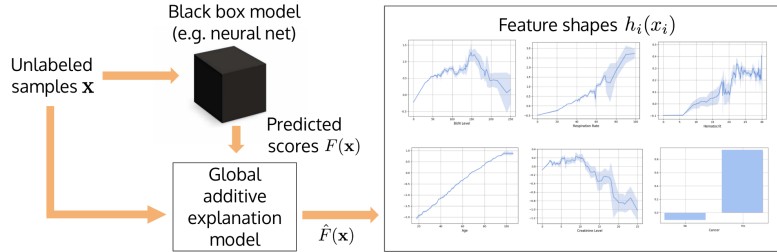

Figure 1: Given a black box model and unlabeled samples (new unlabeled data or training data with labels discarded), our approach leverages model distillation to learn feature shapes that describe the relationship between input features and model predictions.

- We perform a quantitative comparison of feature shapes to other *global* explanation methods in terms of fidelity to the model being explained, accuracy on independent test data, and interpretability through a user study.

## 2  LEARNING GLOBAL ADDITIVE EXPLANATIONS

Although our approach of using model distillation with powerful additive models of the form in equation 1 is new, our work is based on two previous research threads: (1) *decomposing* $F$ into additive $\hat{F}$ to understand how $F$ is affected by its inputs (e.g. Hooker (2007)), and (2) *learning* an interpretable model (often some form of decision tree) to mimic $F$ (e.g. Craven & Shavlik (1995)).

### 2.1  ADDITIVE $\hat{F}$

Global additive explanations have been used to analyze inputs to complex, nonlinear mathematical models and computer simulations (Sobol, 2001), analyze how hyperparameters affect the performance of machine learning algorithms (Hutter et al., 2014), and decompose prediction functions into lower-dimensional components (Hooker, 2004). They are determined by the choice of metric $L$ between $F$ and its approximation $\hat{F}$, degree $d$ of highest order components ($d = 3$ in equation 1, and type of base learner $h$. One common theme of these methods is that they *decompose* $F$ into $\hat{F}$ using numerical or computational methods (e.g. matrix inversion, quasi Monte Carlo). Rather than approximately decomposing $F$ (which can be prohibitively expensive with large $n$ or $p$)[1], we propose to *learn* $\hat{F}$ using model distillation. This is equivalent to choosing $L$ that minimizes the empirical risk between the prediction function $F$ and our global additive explanation $\hat{F}$ on the training data.

To minimize $||F - \hat{F}||_L$, we select two flexible, nonparametric base learners for $h$: splines (Wood, 2006) and bagged trees. This gives us two global additive explanation models: **Student Bagged Additive Boosted Trees (SAT)** and **Student Additive Splines (SAS)**. Other choices of $h$ are possible. We describe our distillation setup to learn these models in Section 2.2. In most of this paper, $\hat{F}$ consists of main components $h_i$ ($d = 1$ in equation 1). Higher order components $h_{ij}$ and $h_{ijk}$ can increase the accuracy of $\hat{F}$, but make interpretation more difficult. When $\hat{F}$ consists of only main components $h_i$, any pairwise or higher order interactions in $F$ are expressed as a best-fit additive approximation added to main components $h_i$, plus a pure-interaction residual. We show examples of this expression in Section 4.1, and show the utility of adding higher order components $h_{ij}$ and $h_{ijk}$, when present in $F$, in Section D.2.

### 2.2  LEARNING $\hat{F}$ USING MODEL DISTILLATION

Neural nets and other black-box models have been approximated by interpretable models such as trees, rule lists, etc., either via model distillation/compression (Craven & Shavlik, 1995; Che et al., 2016; Frosst & Hinton, 2017) or model extraction (Fu, 1994; Sanchez et al., 2015; Lakkaraju et al., 2017; Bastani et al., 2017; Ribeiro et al., 2018). However, all of them approximated classifier models; there has been less work approximating regression models. Another gap in the literature is rule lists for regression: state-of-the-art rule lists (Letham et al., 2015; Angelino et al., 2017) or rule sets (Lakkaraju et al., 2016) do not have regression implementations. Model distillation requires only that the teacher model label a training set, not repeated probing or access to the teacher's internal

---

[1]E.g. Hooker (2007)'s decomposition procedure requires computing $X^T X$, where $X$ is an $n$ by $p$ feature matrix and solving a system of equations involving $2^{p+1}$ samples.

structure or derivatives. This, combined with the applicability of generalized additive models to both classification and regression, means that our approach can approximate a broad class of classification and regression models. We also show in Sections 4.2.2 and 4.3, with a user study, that additive explanations have advantages over decision trees when it comes to interpretability.

**Training teacher neural nets.** Our teacher models are fully-connected neural nets (FNNs) with ReLU nonlinearities. We use the Adam optimizer (Kingma & Ba, 2015) with Xavier initialization (Glorot & Bengio, 2010) and early stopping based on validation loss. At each depth, we search for optimal hyperparameters (number of hidden units, learning rate, weight decay, dropout probability, batch size, enabling batch norm) based on average validation performance on multiple train-val splits and random initializations. The most accurate nets we trained are FNNs with 2-hidden layers and 512 hidden units per layer (2H-512,512); nets with three or more hidden layers had lower training loss, but did not generalize as well. In some experiments we also use a restricted-capacity model with 1 hidden layer of 8 units (1H-8) to compare explanations.

**Training student additive explanation models.** To train SAT and SAS, we find optimal feature shapes $\{h_i\}_1^p$ that minimize the mean square error between the teacher $F$ and the student $\hat{F}$, i.e.

$$L(h_0, h_1, \ldots, h_p) = \frac{1}{T} \sum_{t=1}^{T} \|F(x^t) - \hat{F}(x^t)\|_2^2 = \frac{1}{T} \sum_{t=1}^{T} \|F(x^t) - (h_0 + \sum_{i=1}^{p} h_i(x_i^t))\|_2^2, \quad (2)$$

where $F(x)$ is the output of the teacher model (scores for regression tasks and logits for classification tasks), $T$ is the number of training samples, $x^t$ is the t-$th$ training sample, and $x_i^t$ is its i-$th$ feature. The exact optimization details depend on the choice of $h$. For trees we use cyclic gradient boosting (Buhlmann & Yu, 2003; Lou et al., 2012) which learns the feature shapes in a cyclic manner. As trees are high-variance, low-bias learners (Hastie et al., 2001), when used as base learners in additive models, it is standard to bag multiple trees (Lou et al., 2012; 2013; Caruana et al., 2015). We follow that approach here. For splines, we use cubic regression splines trained using penalized maximum likelihood in R's `mgcv` library (Wood, 2011) and cross-validate the splines' smoothing parameters.

## 2.3 Interpretable Building Blocks of $\hat{F}$: Feature shapes

Our global additive explanation models, SAT and SAS, can be visualized as feature shapes (Figure 1). These are plots with the x-axis being the domain of input feature $x_i$ and the y-axis being the feature's contribution to the prediction $h_i(x_i)$. This way of representing the relationship between input features and model predictions has precedence in interpretability, from work that learned monotonic (Gupta et al., 2016) or concave/convex (Pya & Wood, 2015) feature shapes from original data (i.e. without distillation), to post-hoc explanations such as partial dependence (Friedman, 2001), and Shapley additive explanations dependence plots (Lundberg & Lee, 2017). The latter two are hence natural baselines for SAT and SAS, and we describe the results from our comparison in Section 4.2.1. In Section 4.3, we also describe the results of a user study to evaluate the interpretability of feature shapes, showing that humans are able to understand and use feature shapes.

**How are feature shapes different from feature attribution?** A classic way to interpret black-box models is feature attribution/importance measures. Examples include permutation-based measures (Breiman, 2001), gradients/saliency (see Montavon et al. (2017) or Ancona et al. (2018) for a review), and measures based on variance decomposition (Iooss & Lemaitre, 2015), game theory (Datta et al., 2016; Lundberg & Lee, 2017), etc. We highlight that *feature shapes are different from and more expressive than feature attributions*. Feature attribution is a single number describing the feature's contribution to either the prediction of one sample (local) or the model (global), whereas our feature shapes describe the contribution of a feature, *across the entire domain of the feature*, to the model. Nonetheless, feature attribution, both global and local, can be automatically derived from feature shapes: global feature attribution by averaging feature shape values at each unique feature value; local feature attribution by simply taking one point on the feature shape. In Section 4.3 we show that humans are able to derive feature attribution from feature shapes.

## 3 Evaluating Global Explanations

Lundberg & Lee (2017) suggested the perspective of viewing an explanation of a model's prediction as a model itself. With this perspective, we propose to *quantitatively evaluate explanation models as if they were models*. Specifically, we evaluate not just fidelity (how well the explanation matches

the teacher's predictions) but also accuracy (how well the explanation predicts the original label). Note that Lundberg & Lee (2017) and Ribeiro et al. (2016) evaluated local fidelity (called local accuracy by Lundberg & Lee (2017)), but not accuracy. A similar evaluation of global accuracy was performed by Kim et al. (2016) who used their explanations (prototypes) to classify test data. In our case, we use the feature shapes generated by our approach to predict on independent test data.

**Baselines.** We compare to two types of baselines: (1) additive explanations obtained by querying the neural net (i.e. without distillation): partial dependence, Shapley additive explanations (Lundberg & Lee, 2017) and linearization through gradients; (2) interpretable models learned by distilling the neural net: trees, rules, and sparse linear models.

**Partial dependence** (PD) is a classic global explanation method that estimates how predictions change as feature $x_j$ varies over its domain: $PD(x_j = z) = \frac{1}{T} \sum_{t=1}^{T} F((x_1^t, \ldots, x_j^t = z, \ldots, x_p^t)$ where the neural net is queried with new data samples generated by setting the value of their $x_j$ feature to $z$, a value in the domain of $x_j$. Plotting $PD(x_j = z)$ by $z$ returns a feature shape.

**Linearization through gradient approximation** (GRAD). We construct the additive function $G$ through the Taylor decomposition of $F$, defining $G(x) = F(0) + \sum_{i=1}^{p} \frac{\partial F(x)}{\partial x_i} x_i$, and defining the attribution of feature $i$ of value $x_i$ as $\frac{\partial F(x)}{\partial x_i} x_i$. This formulation is related to the "gradient*input" method (e.g. Shrikumar et al. (2017)) used to generate saliency maps for images.

**Shapley additive explanations** (SHAP). SHAP is a state-of-the-art local explanation method that satisfies several desirable local explanation properties (Lundberg & Lee, 2017). Given a sample and its prediction, SHAP decomposes the prediction additively between features using a game-theoretic approach. We use the python package by the authors of SHAP.

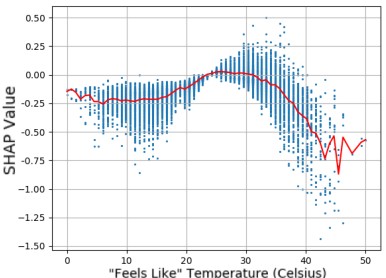

Both GRAD and SHAP provide local explanations that we adapt to a global setting by averaging the generated local attributions at each unique feature value. For example, the global attribution for feature "Temperature" at value 10 is the average of local attribution "Temperature" for all training samples with "Temperature=10". This is the red line passing through the points in Figure 2. Apply-

Figure 2: From SHAP to gSHAP. Blue points are individual SHAP values; red line is gSHAP feature shape.

ing this procedure to GRAD and SHAP's local attributions, we obtain global attributions **gGRAD** and **gSHAP** that we can now plot as feature shapes.

## 4 EXPERIMENTAL RESULTS

First, we validate our approach on synthetic data with known ground-truth feature shapes (Section 4.1). Next, we quantitatively evaluate our approach on real data against other non-distilled additive explanations (Section 4.2.1) and distilled, not-necessarily additive, interpretable models (Section 4.2.2). Third, we design a user study to evaluate the interpretability of feature shapes (Section 4.3). Finally, we further validate our approach with controlled experiments on real data (Section 4.4).

### 4.1 VALIDATION USING SYNTHETIC DATA WITH KNOWN GROUND-TRUTH

For this experiment, we simulate data from synthetic functions with *known* ground-truth feature shapes to see if our approach can recover these feature shapes. We are particularly interested in observing how predicted feature shapes differ for neural nets of different capacity trained on the same data. Our expectation is that for neural nets that are accurate, our predicted shapes would match the ground-truth feature shapes, independent of how the features are used internally by the net. On the other hand, predicted shapes of less accurate neural nets should less accurately match ground-truth shapes.

**Experimental setup.** We designed an additive, highly nonlinear function combining components from synthetic functions proposed by Hooker (2004), Friedman & Popescu (2008) and Tsang et al. (2018): $F_1(\mathbf{x}) = 3x_1 + x_2^3 - \pi^{x_3} + \exp(-2x_4^2) + \frac{1}{2+|x_5|} + x_6 \log(|x_6|) + \sqrt{2|x_7|} + \max(0, x_7) + x_8^4 + 2\cos(\pi x_8)$. Like Tsang et al. (2018), we set the domain of all features to be $\mathcal{U}(-1, 1)$. Like

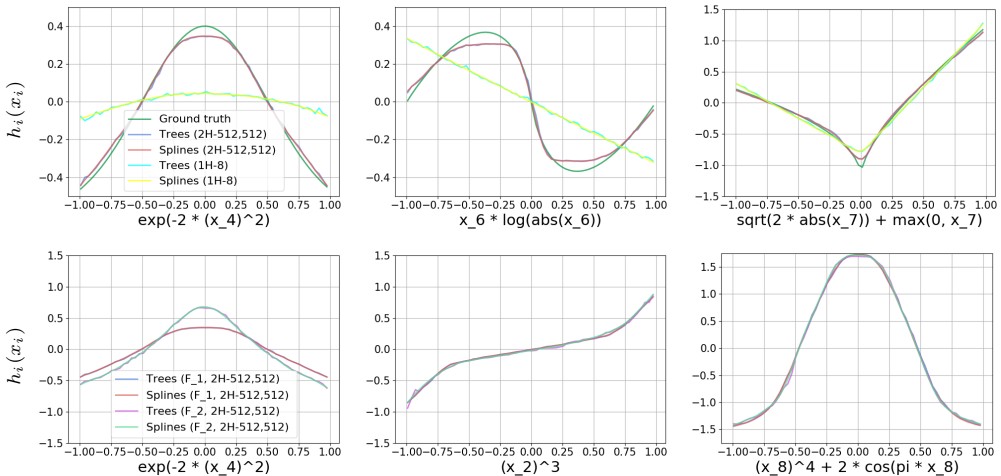

Figure 3: Three of ten feature shapes learned for $F_1$ (top row) and $F_2$ (bottom row) by small (1H-8) and large (2H-512,512) nets. To view shapes learned for all features, see the Appendix.

Friedman & Popescu (2008), we add noise features to our samples that have no effect on $F_1(x)$ via two noise features $x_9$ and $x_{10}$. We trained two teacher neural nets, 2H-512,512 and 1H-8, as described in Section 2.2 to predict $F_1$ using all ten features.

**Performance of teachers and students.** The high-capacity 2H neural net obtained test RMSE of $0.14$, while the low-capacity neural net obtained test RMSE of $0.48$, more than 3x larger. For each neural net, we used our approach to generate two global additive explanation models, SAT and SAS. These explanation models are faithful: the reconstruction RMSE of SAT is $0.14$ for the 1H model and $0.08$ for the 2H model, while the reconstruction RMSE of SAS is $0.14$ for the 1H model and $0.07$ for the 2H model. This suggests that both student methods should accurately represent the teacher, and that they will probably be very similar to each other.

**Do SAT and SAS explain the teacher model, or just the original data?** The top row of Figure 3 compares the feature shapes of our global explanation models SAT and SAS to function $F_1$'s analytic ground-truth feature shapes. SAT and SAS' feature shapes are almost identical. More importantly, it is clear that the feature shapes for the 2H model are different from shapes for the 1H model, and that the shapes for the 2H model better match ground-truth shapes. In general, the shapes of the 2H model are very faithful to the ground-truth shapes, but sometimes fall short when there are sharp changes in the

| Model | All | Agree | Disagree |
|---|---|---|---|
| 1H-8 | 0.483 | 0.407 | 0.489 |
| 2H-512,512 | 0.142 | 0.141 | 0.180 |

Table 1: RMSE error of the teacher models on all samples, compared to the error on samples sampled from regions where the predicted feature shapes "agree" or "disagree" with the ground truth shape.

ground-truth, highlighting the limitations of a 2-hidden-layer neural net (which achieves $0.14$ test RMSE, as noted before). On the other hand, both SAT and SAS' feature shapes for the 1H neural net show a less accurate teacher model that captures the gist of the ground-truth function but not its details, which is consistent with the original teacher RMSE of $0.48$. This shows that our methods fit what the teacher model has learned, and not the original data, and that when the teacher model is accurate the learned shapes match the ground-truth shapes.

**Do SAT and SAS' feature shapes match the real behavior of the model?** To further validate this we use the feature shapes to predict which samples will be inaccurately predicted by the teacher model. Specifically, we sample testing points from the space regions where the predicted feature shapes agree (or disagree) with the the feature shape ground truth (for example, for the 2H model, $x_4 \approx 0$, $x_7 \approx 0$, and $|x_6| \approx 0.3$ define a region where the predicted feature shapes and the ground truth feature shapes disagree) and evaluate them using the teacher model. If the learned feature shapes correctly represent the teacher model, we would expect a lower teacher error on the samples drawn from areas of agreement, and a higher teacher error on the samples drawn from areas of disagreement, compared to the RMSE of all samples. Indeed, as shown in Table 1, points sampled on the agreement regions have lower error than points sampled from the disagreement regions. We performed a two-sample t-test to test if the errors of the samples in the (disjoint) agree and disagree

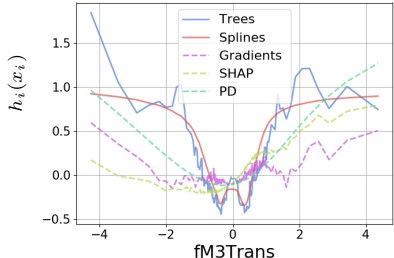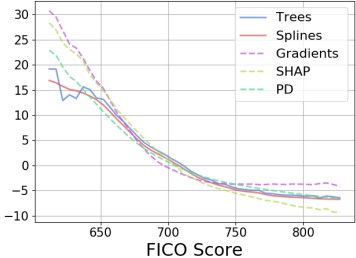

Figure 4: Example feature shapes from Magic (left), and Loan (right). SAT and SAS tend to agree. Best seen on a screen.

groups are significantly different (p-values 4.3e-21 for 1H, 2.3e-4 for 2H). Additionally, to be robust against potential violation of the t-tests normal distribution assumption, we also performed a non-parametric Mann-Whitney-Wilcoxon rank sum test (p-values 5.4e-14 for 1H, 1.8e-6 for 2H). Hence, the difference between the errors is statistically significant, supporting our conclusion that teacher error is higher for samples where feature shapes do not match ground truth, and vice versa, i.e., feature shapes correctly represent the behavior of the models.

**How do interactions between features affect feature shapes?** We design an augmented version of $F_1$ to investigate how interactions in the teacher's predictions are expressed by feature shapes: $F_2(\mathbf{x}) = F_1(\mathbf{x}) + x_1 x_2 + |x_3|^{2|x_4|} + \sec(x_3 x_5 x_6)$. We again simulate 50,000 samples. Note that this function is much harder to learn (the 2H model obtained an RMSE of $0.21$) and also harder for students that do not model interactions to mimic (SAT and SAS obtain fidelity RMSEs of $0.35$). The bottom row of Figure 3 displays features with interactions ($x_4$, $x_2$) and a feature without interactions ($x_8$), and compares them with the shapes from $F_1$. For $x_4$ the part of the interactions that can be approximated additively by $h_i$'s has been absorbed into the $h_i$ feature shapes, changing their shapes as expected. On the other hand, we were still able to recover perfectly the feature shapes of features without interactions (e.g. $x_8$). An interesting case is $x_2$, where, despite interacting with $x_1$, its feature shape has not changed. This is less surprising if we recall that feature shapes describe the *expected importance* of the feature, learned in a data-driven fashion. The interaction term is $x_1 x_2$, which, for $x_1 \sim \mathcal{U}(-1, 1)$, has an expected value of zero, and therefore does not affect the feature shape. Similarly, the expected value of $|x_3|^{2|x_4|}$ when $x_3 \sim \mathcal{U}(-1, 1)$ is $1/(2|x_4| + 1)$, an upward pointing cusp, which modifies the feature shape as shown in Figure 3 (bottom left figure).

## 4.2 QUANTITATIVE COMPARISON OF EXPLANATION METHODS ON REAL DATA

We selected five data sets: two UCI data sets (Bikeshare and Magic), a Loan risk scoring data set from an online lending company (Lending-Club, 2011), the 2018 FICO Explainable ML Challenge's credit data set (FICO, 2018), and the pneumonia data set analyzed by Caruana et al. (2015). Table 2 provides details about the data sets and performance of the 1H and 2H neural nets. 2H neural nets exhibited the most gain in accuracy over 1H neural nets on Bike-share, Loan, and Magic. For the rest of this section we focus on 2H neural nets; results for 1H neural nets are in the Appendix.

|          |        |    |       |      | Performance | |
|----------|--------|----|-------|------|------|------|
| **Data** | $n$    | $p$ | **Type** |    | 1H   | 2H   |
| Bikeshare | 17,000 | 12 | Reg  | RMSE | 0.60 | 0.38 |
| Loan      | 42,506 | 22 | Reg  | RMSE | 2.71 | 1.91 |
| Magic     | 19,000 | 10 | Class | AUC | 92.52 | 94.06 |
| Pneumonia | 14,199 | 46 | Class | AUC | 81.81 | 82.18 |
| FICO      | 9,861  | 24 | Class | AUC | 79.08 | 79.37 |

Table 2: Performance of neural net teachers. For RMSE, lower is better. For AUC, higher is better.

### 4.2.1 COMPARISON WITH NON-DISTILLED ADDITIVE EXPLANATIONS

Table 3 presents the fidelity and accuracy results for SAT and SAS compared to other additive explanations. SAT and SAS yield similar results in all cases, both in terms of accuracy and fidelity. In some cases, such as Magic, SAT (which uses tree base learners) is more accurate, while in some others such as FICO, SAS (which uses spline base learners) has the edge. Trees are locally adaptive smoothers (Breiman et al., 1984) better able to adapt to sudden changes in input-output relationships than splines, but that also gives them more capacity to overfit. We also see this in the feature shapes, where trees tend to be more jagged than splines, particularly in regions with fewer points.

| **Accuracy**
Global Explanation | Bikeshare
RMSE | Loan score
RMSE | Magic
AUC | Pneumonia
AUC | FICO
AUC |
|---|---|---|---|---|---|
| SAT | $0.98 \pm 0.00$ | $2.35 \pm 0.01$ | $90.75 \pm 0.06$ | $82.24 \pm 0.05$ | $79.42 \pm 0.04$ |
| SAS | $0.98 \pm 0.00$ | $2.34 \pm 0.00$ | $90.58 \pm 0.02$ | $82.12 \pm 0.04$ | $79.51 \pm 0.02$ |
| gGRAD | $1.25 \pm 0.00$ | $6.04 \pm 0.01$ | $80.95 \pm 0.13$ | $81.88 \pm 0.05$ | $79.28 \pm 0.02$ |
| gSHAP | $1.02 \pm 0.00$ | $5.10 \pm 0.03$ | $88.98 \pm 0.05$ | $82.31 \pm 0.03$ | $79.36 \pm 0.01$ |
| PD | $1.00 \pm 0.00$ | $4.31 \pm 0.00$ | $82.78 \pm 0.00$ | $82.15 \pm 0.00$ | $79.47 \pm 0.00$ |
| **Fidelity**
Global Explanation | Bikeshare
RMSE | Loan score
RMSE | Magic
RMSE | Pneumonia
RMSE | FICO
RMSE |
| SAT | $0.92 \pm 0.00$ | $1.74 \pm 0.01$ | $1.78 \pm 0.00$ | $0.35 \pm 0.00$ | $0.15 \pm 0.00$ |
| SAS | $0.92 \pm 0.00$ | $1.71 \pm 0.00$ | $1.75 \pm 0.00$ | $0.35 \pm 0.00$ | $0.14 \pm 0.00$ |
| gGRAD | $1.20 \pm 0.00$ | $5.93 \pm 0.01$ | $2.93 \pm 0.01$ | $0.43 \pm 0.00$ | $0.16 \pm 0.00$ |
| gSHAP | $0.96 \pm 0.00$ | $4.83 \pm 0.00$ | $2.15 \pm 0.00$ | $0.46 \pm 0.00$ | $0.16 \pm 0.00$ |
| PD | $0.94 \pm 0.00$ | $3.85 \pm 0.00$ | $3.17 \pm 0.00$ | $0.47 \pm 0.00$ | $0.16 \pm 0.00$ |

Table 3: Accuracy and fidelity of global additive explanations for 2H neural nets. Accuracy is in terms of RMSE for regression tasks and AUROC for classification tasks; fidelity is always RMSE between the student's predictions and the teacher's scores or logits (equation 2). Results for 1H-8 neural nets in Appendix.

Figure 4 displays selected feature shapes for Magic and Loan. The feature shapes produced by PD tend to be much too smooth, which hurts its fidelity and accuracy. Second, in all cases, trees and splines have similar feature shapes and obtain equal or better accuracy and fidelity than the other methods. This is not surprising as the other methods are either local methods adapted to the global setting (gSHAP, gGRAD), or are global explanations that are not optimized to learn the teacher's predictions (PD). For reference, gSHAP when used as a local method (i.e. individual SHAP values, not global feature shapes) achieved a lower RMSE of 0.37 compared to 1.02 on Bikeshare, and a lower RMSE of 1.99 compared to 5.10 on Loan, which is comparable to its 2H teacher's RMSE on test data (Table 2). Hence, methods such as gSHAP excel at local explanations and should be used for those, but, to produce global explanations, global model distillation methods optimized to learn the teacher's predictions perform better.

### 4.2.2 COMPARISON WITH OTHER DISTILLED INTERPRETABLE MODELS

Figure 5 presents the fidelity of SAT measured with RMSE (accuracy has similar pattern) compared to two other distilled interpretable models: decision trees (DT) and sparse L1-regularized linear model (SPARSE), both trained using scikit-learn. We present results as a function of a model-specific parameter $K$ that controls the complexity of the model. For DT, $K$ represents depth, while for SPARSE it represents the number of features with non-zero coefficients. For trees, true model complexity falls

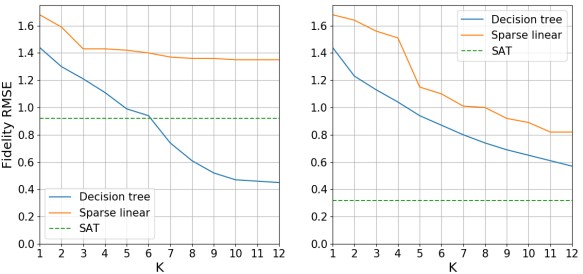

Figure 5: Fidelity (RMSE) of SAT compared to other interpretable models on Bikeshare (left) and Pneumonia (right), as a function of model-specific parameter $K$.

between $K$ and $2^K$ because a binary tree of depth $K$ has $2^K$ leaves ($2^K$ rules), but the complexity is somewhat less than $2^K$ because there is overlap in the rules resulting from the tree structure.

SPARSE obtained by far the worst results in terms of accuracy and fidelity: even if it is interpretable, linear models do not have the fidelity necessary to accurately represent most teacher models. Note that two explanation methods that use sparse linear models (Ribeiro et al., 2016) and rules (Ribeiro et al., 2018) use them as local (not global) explanations, and only for classification (not regression). Trees start to match the accuracy of SAT on Bikeshare at depth $K = 6$ (64 leaves) (Figure A7). However, the largest tree that is readable on letter-size paper has depth $K = 4$ (16 leaves). As seen in the user study in Section 4.3, depth hinders the interpretability of trees. Furthermore, they do not always perform as well as powerful additive models such as SAT. For example, on Pneumonia, a depth $K = 12$ tree ($4,096$ leaves) achieved an accuracy of $80.9$ AUC and a fidelity of $0.57$ RMSE, significantly worse than SAT's $82.24$ AUC and $0.35$ RMSE.

| Task | First stage (n=24) | | Second stage (n=14) | | Third stage (n=12) | |
|---|---|---|---|---|---|---|
| | **SAT-5** | **DT-4** | **SAT-2** | **DT-2** | **SPARSE** | **S-RULES** |
| Rank top 2 features | 75% | 58% | 100% | 85.7% | 83.3% | 0% |
| Rank all (5) features | 45% | 0% | N/A | N/A | N/A | 0% |
| Described increased demand during rush hour | 42% | 0% | 29% | 0% | 0% | 33% |
| Described increased demand during mornings and afternoons | 33% | 0% | 29% | 0% | 0% | 33% |
| Compute change in prediction when feature changes | 33% | 25% | 14% | 100% | 83% | 0% |
| Caught data error | 33% | 8% | N/A | N/A | N/A | 0% |
| Time taken (minutes) | $11.7 \pm 5.8$ | $17.5 \pm 14.8$ | $7.2 \pm 3.2$ | $6.2 \pm 2.2$ | $5.2 \pm 3.1$ | $14.9 \pm 8.4$ |

Table 4: Quantitative results from user study. Since SAT-2, DT-2, and SPARSE only had two features, the task to rank five features does not apply. Since the data error only appeared in the output of SAT-5, DT-4, and S-RULES, the other subjects could not have caught the error.

We tried to compare to rule lists, however, as noted in Section 2.2, state-of-the-art rule lists (Letham et al., 2015; Angelino et al., 2017) do not support regression which is needed for distillation. Hence we used a subgroup discovery algorithm (Atzmueller & Lemmerich, 2012) that supports regression but does not generate disjoint rules. For the rest of this paper we call them S-RULES, short for subgroup rules. Although S-RULES generated semantically meaningful rules, they were no more faithful than SPARSE on Bikeshare (1.42 RMSE), and less faithful than SPARSE on Pneumonia (perhaps because Pneumonia is highly imbalanced).

### 4.3 INTERPRETABILITY EVALUATION WITH HUMAN SUBJECTS

We now describe the results from a user study to see if feature shapes can be understood and used by humans, comparing them to other interpretable models (DT, SPARSE, S-RULES). Table 4 presents quantitative results from the user study.

**Study design.** 50 subjects were recruited to participate in the study. These subjects – STEM PhD students, or college-educated individuals who had taken a machine learning course – were familiar with concepts such as if-then-else structures (for trees and rule lists), reading scatterplots (for SAT), and interpreting equations (for sparse linear models). Each subject only used one explanation model (between-subject design) to answer a set of questions (see Section C) covering common inferential and comprehension tasks on machine learning models: (1) Rank features by importance; (2) Describe relationship between a feature and the prediction; (3) Determine how the prediction changes when a feature changes value; (4) Detect an error in the data.

The study proceeded in three stages. First, we compared the two most accurate and faithful students of the Bikeshare 2H neural net: trees and SAT. We used the depth 4 tree (16 leaves), the largest tree that is readable on letter-size paper, and which does not lag too far behind the depth 6 tree in accuracy (RMSE: SAT 0.98, DT-6 1, DT-4 1.16). DT-4 used five features: Hour, Year, Temperature, Working Day, Season (Figure A4), hence we select the corresponding five feature shapes to display for SAT (Figure A3). In the first stage, 24 of 50 subjects were randomly assigned to see output from DT-4 or SAT-5. In the second stage, we experimented with smaller versions of trees and SAT using only the two most important features, Hour and Temperature. 14 of 50 subjects were randomly assigned to see output from SAT-2 or DT-2. In the last stage, the remaining 12 subjects were randomly assigned to see output from one of the two worst performing models (in terms of accuracy and fidelity): sparse linear models and subgroup-rules.

**Can humans understand and use feature shapes?** From the absolute magnitude of the SAT feature shapes as well as Gini feature importance metrics for the tree, we determined the ground truth feature importance ranking (in decreasing order): Hour, Temperature, Year, Season, Working Day. More SAT-5 than DT-4 subjects were able to rank the top 2 and all features correctly (75% vs. 58%, see Table 4). When ranking all 5 features, 0% of the DT-4 subjects were able to predict the right order, while 45% of the SAT-5 subjects correctly predicted the order of the 5 features, showing that ranking feature importance for trees is actually a very hard task. The most common mistake made by DT-4 subjects (42% of subjects) was to invert the ranking of the last two features, Season

and Working Day, perhaps because Working Day's first appearance in the tree (in terms of depth) was before Season's first appearance (Figure A4).

When asked to describe, in free text, the relationship between the variable Hour and the label, one SAT-5 subject wrote:

> *There are increases in demand during two periods of commuting hours: morning commute (e.g. 7-9 am) and evening commute (e.g. 4-7 pm). Demand is flat during working hours and predicted to be especially low overnight,*

whereas DT-4 subjects' answers were not as expressive, e.g.:

> *Demand is less for early hours, then goes up until afternoon/evening, then goes down again.*

75% of SAT-5 subjects detected and described the peak patterns in the mornings and late afternoons, and 42% of them explicitly mentioned commuting or rush hour in their description. On the other hand, none of the DT-4 subjects discovered this pattern on the tree: most (58%) described a concave pattern (low and increasing during the night/morning, high in the afternoon, decreasing in the evening) or a *positively correlated* relation (42%). Similarly, more SAT-5 subjects were able to precisely compute the change in prediction when temperature changed in value, and detect the error in the data – that spring had lower bike demand whereas winter had high bike demand (bottom right feature shape in Figure A3).

**How do tree depth and number of feature shapes affect human performance?** We also experimented with smaller models, SAT-2 and DT-2, that used only the two most important features, Hour and Temperature. As the models are simpler, some of the tasks become easier. For example, SAT-2 subjects predict the order of the top 2 features 100% of the time (vs 75% for SAT-5), and DT-2 subjects, 85% of the time (vs 58% for DT-4). The most interesting change is in the percentage of subjects able to compute the change in prediction after changing a feature: only 25% for DT-4, compared to 100% for DT-2. Reducing the complexity of the explanation made using it easier, *at the price of reducing the fidelity and accuracy of the explanation*.

Another important aspect is the time needed to perform the tasks: increasing the number of features from 2 to 5 increases the time needed by the subjects to finish the study by 60% for the SAT model, but increases it by 166% for the DT model, that is, interpreting a tree becomes much more costly as the tree becomes deeper (and more accurate), and, in general, subjects make more mistakes. SAT appears to scale up more gracefully.

**Remaining interpretable models: subgroup-rules and sparse linear models.** These explanations were the least accurate and faithful. We found that human subjects can easily read the (few) weights of SPARSE, establish feature importance, and compute prediction changes, and do so quickly – at 5.1 minutes on average, this was the fastest explanation to interpret. However, the model is highly constrained and hid interesting patterns. For example, 100% of the subjects described the relation between demand and hour as increasing, and 83% predicted the exact amount of increase, but none were able to provide insights like the ones provided by SAT-5 and DT-4 subjects.

S-RULES was the second hardest explanation to interpret based on mean time required to answer the questions: 14.9 minutes. Understanding non-disjoint rules appears to be hard: none of the subjects correctly predicted the feature importance order, even for just two features; none were able to compute exactly the change in prediction when feature value changes, and none were able to find the data error. The rules in S-RULES are not disjoint because we could not find a regression implementation of disjoint rules. However, 66% of the subjects discovered the peak during rush hour, as that appeared explicitly in some rules, e.g. "If hour=17 and workingday=yes then bike demand is 5".

To summarize, feature shapes, the interpretable representation we focus on in this paper: (1) allowed humans to perform better (than decision trees, sparse linear models, and rules) at ranking feature importance, pointing out patterns between certain feature values and predictions, and catching a data error; (2) Feature shapes were also faster to understand than big decision trees; (3) Very small decision trees and sparse linear models had the edge in calculating how predictions change when feature values change, but were much less faithful and accurate.

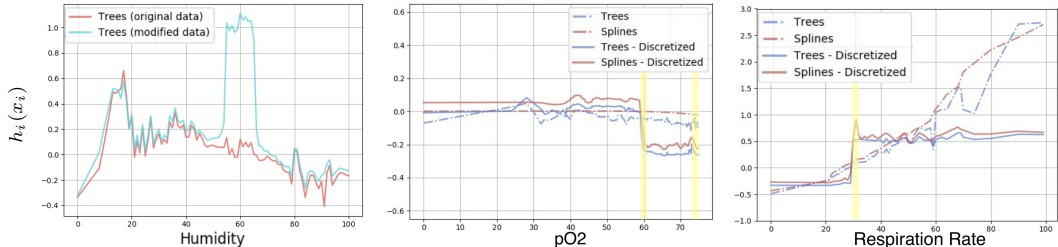

Figure 6: Feature shapes from controlled experiments on real data. Left: Label modification experiment. Center and right: Data modification experiment. See details in Section 4.4.

## 4.4 VALIDATION USING CONTROLLED EXPERIMENTS ON REAL DATA

In this section we further validate global additive explanations on real data. Although here we do not have an analytic solution for the ground-truth feature shapes, we can still design experiments where we modify data in ways that will lead to expected known changes to the ground-truth feature shapes and then verify that these changes are captured in the learned feature shapes.

**Label modification.** On Bikeshare, we added 1.0 to the label (the number of rented bikes) for samples where one of the features (humidity) is between 55 and 65. We then retrained a 2H neural net on the modified data, and applied our approach to learn feature shapes from the 2H net. Ideally, the feature shapes of that new neural net should be almost identical to those of the original net except in that particular range of the humidity feature, where we should see an abrupt "bump" that increases its feature shape value by one. Figure 6 (left) displays the feature shapes. Our method was able to recover the change to the label for the neural net in the new feature shape.

**Data modification: expert discretization.** Sometimes features are transformed before training. For example, in medical data, continuous variables such as body temperature may be discretized by domain experts into bins such as normal, mild fever, moderate fever, high fever, etc. In this experiment we test if our additive explanation models can recover these discretizations from the neural net without access to the discretized features. We train our student additive models using as input features *the original un-discretized features*, but using as labels the outputs of a neural net that was trained on discretized features. Our expectation is that if the student models are an accurate representation of what the neural net learned from the discretized features, they will detect the discretizations, even if they never have access to the discretized features or to the internal structure of the neural-net teacher. We study the feature shapes of two features in the Pneumonia data (Blood $pO_2$ and Respiration Rate) in Figure 6, where we compare the feature shapes learned from teachers trained on the original continuous data (dotted lines) with those from teachers trained on discretized features (solid lines). Recall that in both cases the student models only saw non-discretized features to generate feature shapes. Our approach captures the expected discretization intervals (in yellow) as described in Cooper et al. (1997).

We discuss extensions & applications of our approach in Section D in the Appendix, including visualizing a neural net as it is trained (`https://youtu.be/ATNcgurNHhc`).

## 5 CONCLUSIONS

We presented a method for "opening up" complex models such as neural nets trained on tabular data. The method, based on distillation with high-accuracy additive models, has clear advantages over other approaches that learn additive explanations but not using distillation, and non-additive explanations using distillation. Our global additive explanations do not aim to compete with local explanations or non-additive explanations such as decision trees. Instead, we show that different interpretable representations work well for different tasks, and global additive explanations are valuable for important tasks that require quick understanding of feature-prediction relationships. Although in this paper we focus on explaining FNNs, the method will work with any classification or regression model including random forests and CNNs, but is not designed to work with raw image inputs such as pixels where providing a global explanation in terms of input pixels is not meaningful. One way to address this is to define more meaningful "features", e.g. the intermediate activations of a CNN, which are known to implicitly represent a hierarchy of concepts.

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

APPENDIX

A    ALL FEATURE SHAPES FOR $F_1$ AND $F_2$ SYNTHETIC FUNCTIONS (FROM SECTION 4.1)

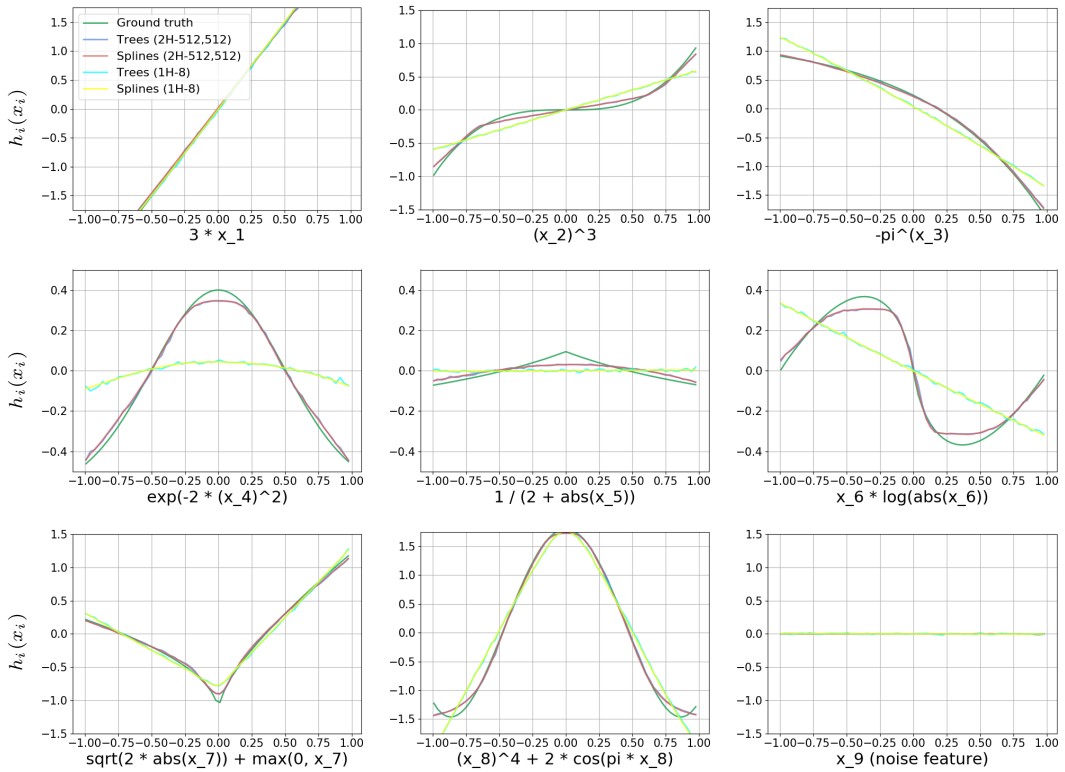

Figure A1: Feature shapes for features $x_1$ to $x_9$ of $F_1$ from Section 4.1. Notice how $x_9$, which is a noise feature that does not affect $F_1$, has been assigned an importance of approximately $0$ throughout its range. The feature shape of $x_{10}$, another noise feature, is very similar to $x_9$ and hence not included here.

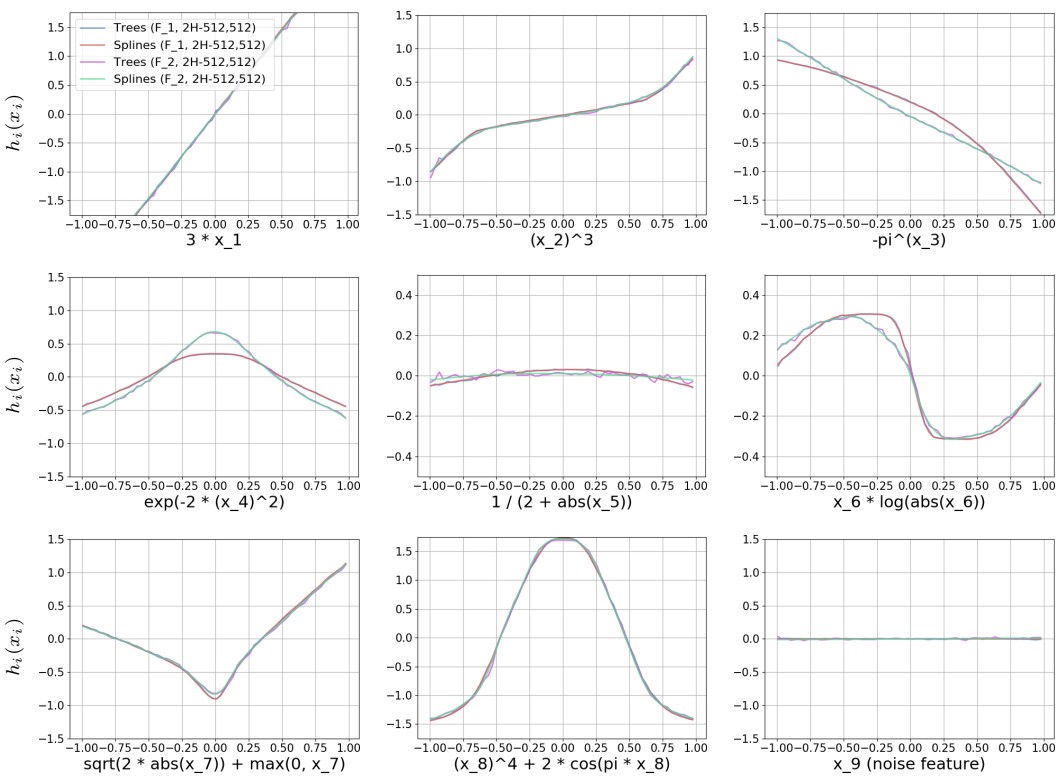

Figure A2: Feature shapes for features $x_1$ to $x_9$ of $F_2$ from Section 4.1. Notice how $x_9$, which is a noise feature that does not affect $F_2$, has been assigned an importance of approximately $0$ throughout its range. The feature shape of $x_{10}$, another noise feature, is very similar to $x_9$ and hence not included here.

## B    Quantitative Comparison of Explanation Methods on Real Data: all neural net teachers (continued from Section 4.2)

| **Accuracy** Teacher | Global Explanation | Bikeshare RMSE | Loan score RMSE | Magic AUC | Pneumonia AUC | FICO AUC |
|---|---|---|---|---|---|---|
| 1H-8 | SAT | $1.00 \pm 0.00$ | $2.82 \pm 0.00$ | $90.44 \pm 0.05$ | $82.01 \pm 0.05$ | $79.43 \pm 0.02$ |
| | SAS | $1.00 \pm 0.00$ | $2.82 \pm 0.00$ | $90.43 \pm 0.03$ | $81.91 \pm 0.06$ | $79.56 \pm 0.02$ |
| | gGRAD | $1.08 \pm 0.00$ | $2.84 \pm 0.00$ | $84.52 \pm 0.67$ | $81.63 \pm 0.06$ | $79.34 \pm 0.05$ |
| | gSHAP | $1.04 \pm 0.00$ | $2.87 \pm 0.00$ | $89.94 \pm 0.03$ | $82.02 \pm 0.02$ | $79.49 \pm 0.02$ |
| | PD | $1.00 \pm 0.00$ | $3.00 \pm 0.00$ | $85.11 \pm 0.00$ | $82.03 \pm 0.00$ | $79.46 \pm 0.00$ |
| 2H-512,512 | SAT | $0.98 \pm 0.00$ | $2.35 \pm 0.01$ | $90.75 \pm 0.06$ | $82.24 \pm 0.05$ | $79.42 \pm 0.04$ |
| | SAS | $0.98 \pm 0.00$ | $2.34 \pm 0.00$ | $90.58 \pm 0.02$ | $82.12 \pm 0.04$ | $79.51 \pm 0.02$ |
| | gGRAD | $1.25 \pm 0.00$ | $6.04 \pm 0.01$ | $80.95 \pm 0.13$ | $81.88 \pm 0.05$ | $79.28 \pm 0.02$ |
| | gSHAP | $1.02 \pm 0.00$ | $5.10 \pm 0.00$ | $88.98 \pm 0.05$ | $82.31 \pm 0.03$ | $79.36 \pm 0.01$ |
| | PD | $1.00 \pm 0.00$ | $4.31 \pm 0.00$ | $82.78 \pm 0.00$ | $82.15 \pm 0.00$ | $79.47 \pm 0.00$ |
| **Fidelity** Teacher | Global Explanation | Bikeshare RMSE | Loan score RMSE | Magic RMSE | Pneumonia RMSE | FICO RMSE |
| 1H-8 | SAT | $0.64 \pm 0.00$ | $1.15 \pm 0.00$ | $1.12 \pm 0.00$ | $0.30 \pm 0.00$ | $0.21 \pm 0.00$ |
| | SAS | $0.64 \pm 0.00$ | $1.14 \pm 0.00$ | $1.11 \pm 0.00$ | $0.30 \pm 0.00$ | $0.21 \pm 0.00$ |
| | gGRAD | $0.71 \pm 0.00$ | $1.54 \pm 0.00$ | $35.40 \pm 4.47^*$ | $0.36 \pm 0.00$ | $0.24 \pm 0.00$ |
| | gSHAP | $0.68 \pm 0.00$ | $1.28 \pm 0.00$ | $1.29 \pm 0.00$ | $0.38 \pm 0.00$ | $0.22 \pm 0.00$ |
| | PD | $0.65 \pm 0.00$ | $1.37 \pm 0.00$ | $1.94 \pm 0.00$ | $0.38 \pm 0.00$ | $0.25 \pm 0.00$ |
| 2H-512,512 | SAT | $0.92 \pm 0.00$ | $1.74 \pm 0.01$ | $1.78 \pm 0.00$ | $0.35 \pm 0.00$ | $0.15 \pm 0.00$ |
| | SAS | $0.92 \pm 0.00$ | $1.71 \pm 0.00$ | $1.75 \pm 0.00$ | $0.35 \pm 0.00$ | $0.14 \pm 0.00$ |
| | gGRAD | $1.20 \pm 0.00$ | $5.93 \pm 0.01$ | $2.93 \pm 0.01$ | $0.43 \pm 0.00$ | $0.16 \pm 0.00$ |
| | gSHAP | $0.96 \pm 0.00$ | $4.83 \pm 0.01$ | $2.15 \pm 0.00$ | $0.46 \pm 0.00$ | $0.16 \pm 0.00$ |
| | PD | $0.94 \pm 0.00$ | $3.85 \pm 0.00$ | $3.17 \pm 0.00$ | $0.47 \pm 0.00$ | $0.16 \pm 0.00$ |

Table A1: Accuracy and fidelity of global explanation models across 1H and 2H teacher neural nets and datasets. Table 3 is a subset of this table with only 2H neural nets.

In general, the lower-capacity 1H neural nets are easier to approximate (i.e. better student-teacher fidelity), but their explanations are less accurate on independent test data. Students of simpler teachers tend to be less accurate even if they are faithful to their (simple) teachers. One exception is the FICO data, where the fidelity of the 2H explanations is better. Our interpretation is that many features in the FICO data have almost linear feature shapes (see Figure A5 for a sample of features), and the 2H model may be able to better capture fine details while being simple enough that it can still be faithfully approximated. The accuracy of the SAT and SAS for 1H and 2H neural nets are comparable, taking into account the confidence intervals.

On the Magic data, the fidelity of the gGRAD explanation to the 1H neural net (see * in Table A1) is markedly worse than other explanation methods. We investigate the individual gradients of the 1H neural net with respect to each feature ($\frac{\partial F(x)}{\partial x_i}$ in GRAD equation in Section 3). 99% of them have reasonable values (between -5.6 and 6). However, 3 are larger than 1,000 (with none between 6 and 1,000) and 13 are lower than -1,000 (with none between -1,000 and -5.6), resulting in the ensuing gGRAD explanation generating extreme predictions for several samples that are not faithful to the teacher's predictions. Because AUC is a ranking loss, accuracy (AUC) is less affected than fidelity (RMSE) by the presence of these extreme values. This shows that gGRAD explanations may be problematic when individual gradients are arbitrarily large, e.g. in overfitted neural nets.

## C  USER STUDY MATERIALS (FROM SECTION 4.3)

All 50 user study subjects answered these questions:

1. What is the most important variable for predicting bike demand?
2. Rank all the variables from most important to least important for predicting bike demand.
3. Describe the relationship between the variable Hour and predicted bike demand.
4. What are variables for which the relationship between the variables and predicted bike demand is positive?
5. The Hour is 11. When Temperature increases from 15 to 20, how does predicted bike demand change?
6. There is one error in the data. Any idea where it might be? Cannot find the error is an ok answer.

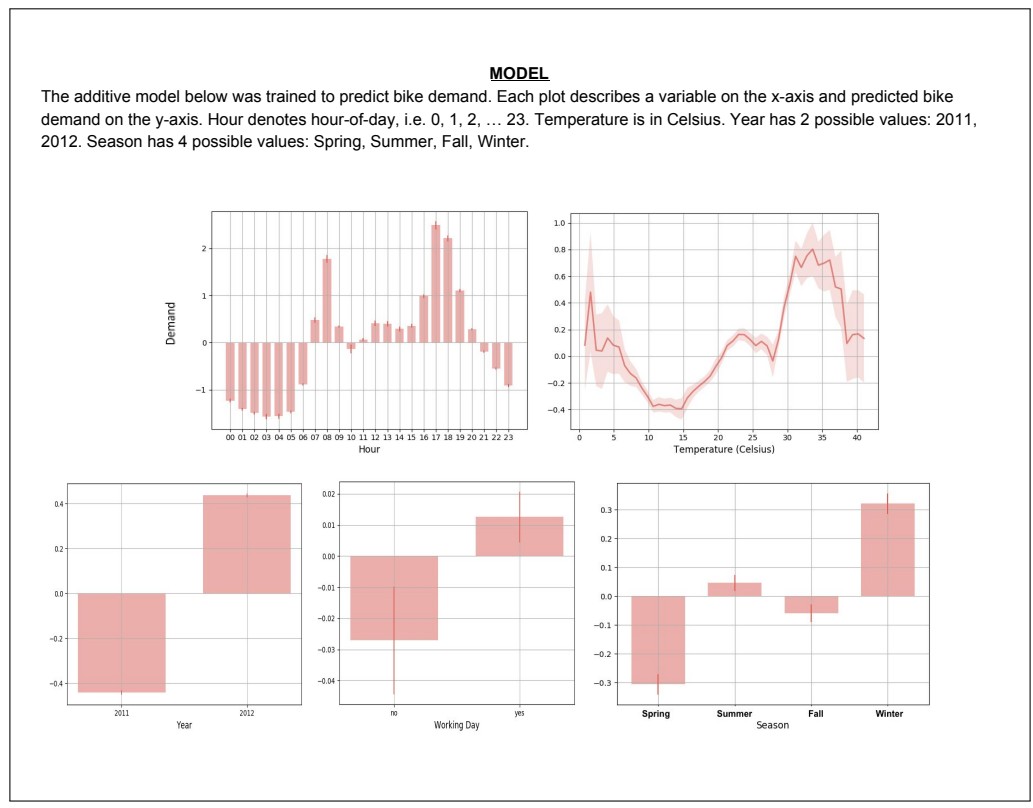

Figure A3: Model output shown to SAT-5 subjects.

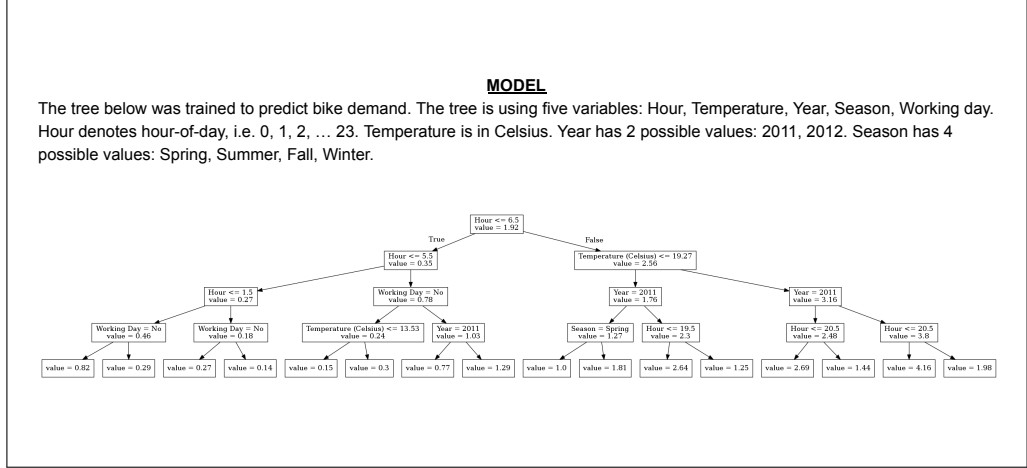

Figure A4: Model output shown to DT-4 subjects. This visual presentation was simplified from the visual presentation in Figure A7, removing the color and number of samples in each node, to improve readability for the user study.

# D APPLICATIONS AND EXTENSION

In this section we discuss applications of our approach and extensions to include higher-order interactions.

## D.1 INSIGHTS FROM GLOBAL ADDITIVE EXPLANATIONS

**Checking for monotonicity.** Domains such as credit scoring have regulatory requirements that prescribe monotonic relationships between predictions and some features (Federal Reserve Governors, 2007). For example, the 2018 FICO Explainable ML Challenge encouraged participants to impose monotonicity on 16 features (FICO, 2018). We use feature shapes to see if the function learned by the neural net is monotone for these features. 15 of 16 features are monotonically increasing/decreasing as required. One feature, however, "Months Since Most Recent Trade Open" was expected to decrease monotonically, but actually increased monotonically. This is true not just in our explanations, but also in PD, gGRAD, and gSHAP (Figure A5). Note that testing for monotonicity requires global explanations or checking and aggregating many local explanations.

With the insight from the global explanations that the neural net may not be exhibiting the expected pattern for "Months Since Most Recent Trade Open", we perform a quick experiment to verify this in the neural net. We sample values of this feature across its domain, set all data samples to this value (for this feature), and obtain the neural net's predictions for these modified samples. The majority of samples (70%) had predictions that increased as this feature increased across its domain, confirming that on average, the neural net exhibits a monotonically increasing instead of decreasing pattern for this feature. Note that we could not have checked for a monotonicity pattern (which is by definition a global behavior) without checking and aggregating multiple local explanations.

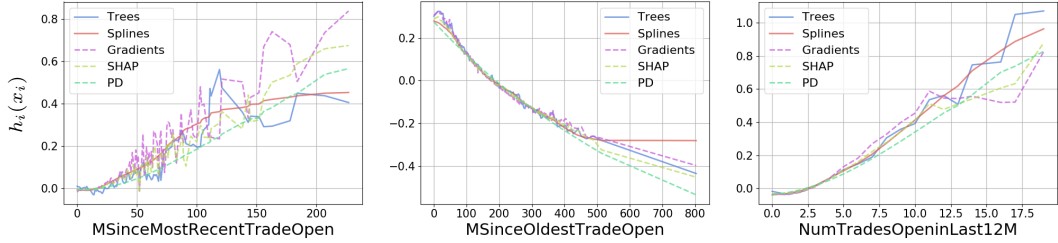

Figure A5: 3 of 16 features with expected monotonically increasing/decreasing patterns in the FICO data. "Months Since Most Recent Trade Open", the leftmost figure, was expected to decrease monotonically, but actually increased monotonically according to all explanations. The two figures on the right are two related features, "Months Since *Oldest* Trade Open" and "Number of Trades Open in Last 12 Months", both of which exhibit the expected monotonically decreasing/increasing patterns.

**Visualizing neural net training: from underfit to overfit.** Using additive models to peek inside a neural net creates many opportunities. For example, we can see what happens in the neural net when it is underfit or overfit; when it is trained with different losses such as squared, log, or rank loss or with different activation functions such as sigmoid or ReLUs; when regularization is performed with dropout or weight decay; when features are coded in different ways; etc. The video at `https://youtu.be/ATNcgurNHhc` shows what is learned by a neural net as it trains on a medical dataset. The movie shows feature shapes for five features before, at, and after the early-stopping point as the neural net progresses from underfit to optimally fit to overfit. We had expected that the main cause of overfitting would be increased non-linearity (bumpiness) in the fitting function, but a significant factor in overfitting appears to be unwarranted growth in the confidence of the model as the logits grow more positive or negative than the early-stopping shape suggests is optimal.

## D.2 Extending $\hat{F}$ to Include Interactions

Functions learned by neural nets cannot always be represented with adequate fidelity by the additive function $\hat{F}$ in equation 1. We can improve $\hat{F}$'s expressive power by adding pairwise and higher-order components $h_{ij}$, $h_{ijk}$, and so on to account for interactions between two or more input features. In Bikeshare, RMSE decreases from 0.98 to 0.60 when we add pairwise interactions to the student model. Figure A6 shows an interesting interaction between two features: "Time of Day", and "Working Day". On working days, the highest bike rental demand occurs at 7-9am and 5-7pm, but on weekends there is very low demand at 7-9am (presumably because people are still sleeping) and at 5-7pm, and demand peaks during midday from 10am-4pm. These two features also form a three-way interaction with temperature. When-

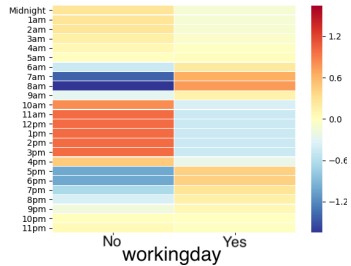

Figure A6: An important pairwise interaction in Bikeshare.

ever the teacher neural net learned these (and other) interactions, a global explanation method must also incorporate interactions if it is to provide high-fidelity explanations of the teacher model. Our approach is able to do so by adding higher-order components $h_{ij}$, $h_{ijk}$, and so on to the global additive explanation $\hat{F}$.

E   TREE THAT MATCHED SAT FIDELITY ON BIKESHARE DATASET (FROM SECTION 4.2.2)

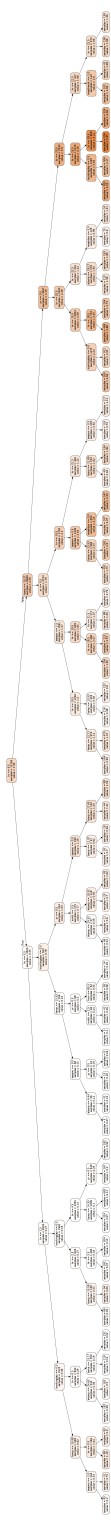

Figure A7: Tree of depth 6 (64 leaves), the least deep tree that matched SAT's fidelity. This uses the default tree visualizer in scikit-learn. For the tree of depth 4 (DT-4) presented in the user study (Figure A4), we removed the color and number of samples in each node to increase readability.

