# OpenReview forum: "Learning Global Additive Explanations for Neural Nets Using Model Distillation"
_ICLR.cc/2019/Conference_

### Official Review · AnonReviewer3 · 2018-11-01
**good paper**

**Rating:** 6
**Confidence:** 4

**Review:**

This paper is of high quality and clarity. I think it's originality is at least decent. Whether it is significant or not depends on how significant one thinks fully connected neural networks are as these are the models for which this explanation model makes sense.

Good things:
- It is a very elegant method. It is also very simple (in a good way).
- The paper is really well written.
- The experiments are carefully conducted and are indeed showing what the authors describe.
- I think the method is potentially of practical use.

Problems:
- I think qualifying this paper as a paper on representation learning is a small stretch. It would be perhaps more suitable to submit it to ICML or NIPS. I think it is close enough though.
- The font is too small in many figures. It is impossible to read it.
- I am not sure whether model compression is actually necessary here. How good is the additive model if it is trained as a standalone model straight from the training data in comparison to the neural networks and to the additive model when trained with model compression? If the neural network and the additive model were similar in performance when trained from scratch, I would not see the point in explaining the neural network.
- Only makes sense to apply this to fully connected networks.

---

> ### Author Response · Authors · 2018-11-10
> **Addressing comments**
>
> We'd like to thank for the reviewer for the overall positive and encouraging comments. We will update the figure fonts to make sure they are readable. Regarding the other concerns:
>
> - "Only makes sense to apply this to fully connected networks". Although our evaluation is on fully-connected neural networks (FNNs), our approach is not technically limited to FNNs. We have applied our approach to other teachers such as gradient boosted trees and random forests, linear models (as a sanity check, because there we expect the student to have linear feature shapes, and we do find that), and even other non-linear additive models. We note that it is not the model class but the data input that determines how informative and practical our feature shapes are. For example, one could apply this approach to a convolutional neural net (CNN) or a FNN trained on MNIST or CIFAR, and the result would be a feature shape for each pixel position, that tells us how much the value at that pixel position contributes to the prediction. Despite being correct, this feature shape would not be as informative as the feature shapes of “tabular” data (common in healthcare and social science domains) that we use in this paper.
>
>
> - “I think qualifying this paper as a paper on representation learning is a small stretch. [...]. I think it is close enough though": We agree that our approach differs somewhat from traditional representation learning tasks. But, along with the referee, we still see this as a form of finding representations. In particular, we are finding the best additive representation of the FNN’s prediction function.
>
>
> - " If the neural network and the additive model were similar in performance when trained from scratch, I would not see the point in explaining the neural network". We agree that if the neural net’s training data (features and labels) are available and the additive model can be trained to be highly accurate and deployed, then we don’t need to explain a neural net using an additive model.  However, this is not always the case. E.g.:
>
> i) Availability of original training data. Data regulations --e.g. General Data Protection Regulation (GDPR) or California's Consumer Privacy Act (CPA)-- may limit how the data can be used or retained. In that case, the data and labels used to train the original model may no longer be available to train the additive model on directly. However, training the additive model to inspect the original model does not require the original data and labels, only a second set of unlabeled data (to be passed through the original model to get soft labels), which is easier to obtain.
>
> ii) Deployment constraints. In some scenarios, the model class may be constrained due to external factors or preferences. For example, a model has to be deployed using specific hardware or have particular requirements in terms of memory bandwidth, CPU, etc. Another example would be a model requiring a costly verification and approval process before it can be deployed. Then, the original model could not be simply replaced by the additive model, even if the additive model could be trained to the same level of accuracy or higher. However, it would still be valuable to have techniques to inspect the model.
>
> For these reasons, we argue that post-hoc explanations of black-box models are still useful, and our work contributes one such approach. If the reviewer has any additional experiments in mind, we would be happy to perform them.

---

> ### Author Response · Authors · 2018-11-26
> **paper updated**
>
> The paper has been updated to demonstrate the interpretability of feature shapes (the interpretable representation of additive models) compared to other interpretable models such as decision trees, sparse linear models, or rules.
>
> We have added 2 sections -- Section 4.2.2 and 4.3 -- to the paper, including a new user study with 50 participants performing common inferential and comprehension tasks on machine learning models. e.g. ranking features by importance, describing relationship between feature and predictions, and detecting an error in the data. Please see Section 4.3 for a full writeup of the user study.
>
> To accommodate the new user study, evaluation against new baselines (decision trees, etc.), and increased plot sizes, we restructured the paper. Besides adding Sections 4.2.2 and 4.3, the “Applications and Extensions” section was moved to the appendix, and the end of Section 1, much of Section 2 and Section 5 was rewritten. A note about the new baselines was also added to Section 3.

---

### Official Review · AnonReviewer2 · 2018-11-12
**Well written paper but method lacks novelty**

**Rating:** 4
**Confidence:** 5

**Review:**

Summary: This paper makes an interesting contribution of providing global explanations of black box models (such as neural nets) using a special class of models called generalized additive models.  While the paper is well written and experiments are quite detailed, I have some problems with some of the basic premises of this work.
1. The concept of using simpler models to approximate other complex models (model distillation) is not new and has been explored quite a bit already in ML literature. The only new proposition of this work is to use generalized additive models to approximate other complex models. This seems rather incremental.
2. The premise behind using generalized additive models (GAMs) to explain other complex models is that GAMs are interpretable. I am not convinced about this premise. While I can intuitively see that GAMs might be able to better approximate complex models compared to rules and trees, I highly doubt if they are even interpretable.

Pros:
1. The paper is well written
2. Experiments are very detailed and thorough

Cons:
1. The proposed approach lacks novelty
2. Experimentation lacks a user study which helps understand if and when GAMs are at least as interpretable as rule-based approaches.

Detailed Comments:
I actually like the way this paper is written and executed. The writing is very clear and experiments are quite thorough. But, as discussed earlier, I have some issues with the basic premises of this paper i.e., novelty of the proposed approach and justification for the claim that GAMs are interpretable. I would encourage the authors to discuss these two aspects in their rebuttal. I would strongly encourage the authors to carry out at least a simple user study which compares the interpretability aspect of GAMs with rule-based or prototype based approaches.

---

> ### Author Response · Authors · 2018-11-26
> **Addressed comments in main thread**
>
> Thanks for your comments and suggestions. Since your concerns are similar to Reviewer 1’s, we addressed them jointly.  Please see our comments Parts 1 and 2 in the main thread, and the updated paper.

---

### Official Review · AnonReviewer1 · 2018-11-12
**well written with thorough experiments, but limited novelty and scope**

**Rating:** 6
**Confidence:** 5

**Review:**

Summary:
This paper incorporates Generalized Additive Models (GAMs) with model distillation to provide global explanations of neural nets (fully-connected nets as black-box in the paper). It is well written with detailed experiments of synthetic and real tabular data, and makes some contribution towards the interpretability of black-box models. However, it lacks novelty and is limited to tabular data as presented.

Pros:
- The paper is well written.
- The experiments are detailed and thorough with both synthetic and real data.

Cons:
- The novelty is limited. The core consists of GAMs well studied in the literature, e.g. Caruana et al 2015. Admittedly, this work also tries to incorporate model distillation to explain black-box models globally. The concept of student models approximating teacher models is not new either. The originality seems incremental in both directions.
- The scope is limited. The paper only presents applications in tabular data. Also, it would be better to experiment with black-box models beside simple fully-connected nets.
- The interpretability is not convincing. It is not sufficient to demonstrate the interpretability of the proposed method, or the expressive advantage of feature shapes. It is encouraged to include studies with human subjective to compare against other existing interpretable approaches.

Specifics:
- With Figure 3, it is not convincing that the student model actually explains the teacher model, so the paper tries to elaborate more with Table 1. I think Table 1 also needs more details to help, such as the significance of error difference and '-' elements.
- Many figures are hard to read mostly because of font, color, and overlap.

---

> ### Author Response · Authors · 2018-11-26
> **Addressed comments in main thread**
>
> Thanks for your comments and suggestions. Since your concerns are similar to Reviewer 2's, we addressed them jointly.  Please see our comments Parts 1 and 2 in the main thread, and the updated paper.

---

### Author Response · Authors · 2018-11-26
**Addressing Reviewer 1 and 2's comments - Part 2**

We again thank the reviewers for their comments and suggestions. Since Reviewers 1 and 2 shared similar concerns, we address them jointly here. This is the second part of our answer.

Scope -> Our paper focuses on tabular data, which is common in many settings. With neural nets experiencing a resurgence for data such as images, speech and raw text, neural nets are also starting to be used more on tabular data where previously one would have used boosted trees or random forests. To give a few examples, a recent paper on tabular electronic medical records [Rajkomar et al., 2018] used a 2-hidden layers, 1024 hidden units per layer fully connected neural net (FNN).  Another recent paper using tabular demographic and education data to predict high school GPA [Davidson, forthcoming] used FNNs with 1-3 hidden layers and up to 256 hidden units per layer. Both used different feature attribution methods to interpret their nets, which points to the need for explanation methods for FNNs on tabular data, and the lack of consensus on how to achieve this.

We note that our approach is not limited to FNNs. As mentioned in our previous answer to Reviewer 3, we have applied our approach to other teachers such as boosted trees, random forests, linear models, etc. as our approach is teacher-model-agnostic.

Regarding tabular data, our approach could be used with other types of data provided meaningful “features” could be defined. As mentioned in the Conclusion section of our paper, our approach is not designed to work with raw inputs such as pixels where providing a additive global explanation in terms of input pixels is not meaningful -- a feature shape for each pixel position simply tells us how much value that pixel position contributes to the prediction, which is not informative. However, if more meaningful “features” (i.e., concepts that apply to multiple images such as textures, colors, regions or object parts) can be defined, then our approach probably could be applied. One approach would be to use the intermediate neural activations of a CNN, which often represent a hierarchy of concepts, as features and compute feature shapes for these convolutional activations. Combined with neuron visualization techniques (e.g. [Bau et al., 2017]), we might then then obtain information about how specific intermediate neurons (and what they represent) contribute to the prediction.

Additional comments:
Clarity of Table 1 -> Thank you Reviewer 1 for pointing out that the presentation of the Table 1 experiment in Section 4.1 could be improved. We have made the goal of the experiment clearer and updated the table to highlight the accuracy differences between samples where our feature shapes match and do not match ground truth. The results of a two-sample t-test and a Mann-Whitney-Wilcoxon test (in case the t-test’s normal distribution assumption is violated) both returned very low p-values, supporting our conclusion that feature shapes correctly represent the behavior of the teacher. Please see Section 4.1 for a full write-up.

Font size and plot size -> We have increased the figure fonts (axes, legends) and increased the size of some figures to improve the overall presentation.  Thanks.

Restructuring and rewriting -> To accommodate the new user study, evaluation against new baselines (decision trees, etc.), and increased plot sizes, we restructured the paper. Besides adding Sections 4.2.2 and 4.3, the “Applications and Extensions” section was moved to the appendix, and the end of Section 1, much of Section 2 and Section 5 was rewritten. A note about the new baselines was also added to Section 3.

References
[Angelino, E. et al., 2017] Learning certifiably optimal rule lists. KDD
[Bau et al., 2017] Network Dissection: Quantifying Interpretability of Deep Visual Representations. CVPR
[Davidson, forthcoming] Black Box Models and Sociological Explanations: Predicting High School GPA Using Neural Networks. Socius
[Hastie & Tibshirani, 1990] Generalized Additive Models. CRC Press.
et al., 1990]
[Lakkaraju et al., 2016] Interpretable decision sets:  A joint
framework for description and prediction. In KDD
[Letham et al., 2015] Interpretable classifiers using rules and bayesian analysis: Building a better stroke prediction model. The Annals of Applied Statistics
[Rajkomar et al., 2018] Scalable and accurate deep learning with electronic health records. npj Digital Medicine
[Ribeiro et al., 2016] “Why Should I Trust You?”: Explaining
the predictions of any classifier. KDD
[Ribeiro et al., 2018] Anchors: High-precision model-agnostic
explanations. AAAI

---

### Author Response · Authors · 2018-11-26
**Addressing Reviewer 1 and 2's comments: Part 1**

We thank the reviewers for their comments and suggestions. Since Reviewers 1 and 2 shared similar concerns, we address them jointly here. This is the first part of our answer.

Interpretability of additive models compared to other interpretable models such as decision trees, sparse linear models, or rules -> We added 2 new sections -- 4.2.2 and 4.3 -- to the paper to demonstrate the interpretability of feature shapes (the interpretable representation of additive models):
- Section 4.2.2 quantitatively evaluates distilled decision tree and sparse linear models in terms of accuracy and fidelity. To summarize, sparse linear models are intelligible but tend to be less faithful and accurate. Decision trees sometimes exhibit better fidelity and accuracy than additive models, but only when they are so deep that they become less intelligible.
- Section 4.3 presents the results of a new user study with 50 participants performing common inferential and comprehension tasks on machine learning models. e.g. ranking features by importance, describing relationship between feature and predictions, and detecting an error in the data. The user study showed that: (1) feature shapes allowed humans to perform better (than decision trees, sparse linear models, and rules) at ranking feature importance, pointing out patterns between certain feature values and predictions, and catching a data error; (2) Feature shapes were also faster to understand than big decision trees; (3) Less complex small decision trees and sparse linear models had the edge in calculating how predictions change when feature values change, but were less faithful and accurate. See Section 4.3 for a full write-up.

Generalized additive models (GAMs) were specifically designed to be interpretable [Hastie et al. 1990], and there is a long history of using feature shape plots in the interpretability literature, from work that learned monotonic or concave/convex feature shapes from original data (i.e., without distillation), to post-hoc explanations such as partial dependence and Shapley dependence plots. Section 2.3 now has more citations to these works.

Novelty -> As pointed out by Reviewer 1 and 2, both additive models and model distillation are, independently, well studied. However, that is not the focus of our work. The main contribution of our work is to show that learning additive explanations through model distillation has clear advantages over alternate approaches that learn additive explanations but not using distillation (e.g. partial dependence, Shapley additive explanations), and non-additive explanations that use distillation (e.g. decision trees). This is because a) unlike non-distillation methods, the loss optimized via model distillation increases the accuracy and fidelity of the learned explanation, and b) additive explanations have clear advantages over other interpretable representations such as rules and decision trees ---  we now show quantitatively, and now supported by a user study, that additive representations are good for tasks such as ranking features by importance (Section 4.3), determining if the relationship between a feature and model predictions is positive (Section 4.3) and/or monotonic (Section D.1), and detecting changes in feature-prediction relationships as a model trains (Section D.2). Imagine, for example, how uninterpretable the neural net movie in Section D.2 would be if we created a movie using trees or rules instead of feature shapes -- seeing the difference between two trees is difficult because trees do not always change smoothly the way feature shapes do.

In addition, most previous work, whether using model distillation or model extraction (see Section 2.2 for citations to these works) to approximate black-box models focus on classifier models; there has been little work in approximating regression models. Also, state-of-the-art rule lists [Letham et al., 2015; Angelino et al., 2017] or rule sets [Lakkaraju et al., 2016] only have classification implementations, not regression. Model distillation requires only that the teacher model label a training set, not repeated probing or access to the teacher's internal structure or derivatives.  This, combined with the applicability of generalized additive models to both classification and regression, means that our approach can approximate a broad class of classification and regression models. Hence, our paper contributes to the scarce literature on approximating regression models for interpretability.

---

### Meta-Review · Area_Chair1 · 2018-12-18
**Limited scope and novelty**

**Confidence:** 4
**Recommendation:** Reject

**Metareview:**

This paper introduces a distillation approach for black-box classifiers that trains generalized additive models (GAM), an additive model over feature shapes, thus providing global explanations for the model. Given the importance of interpretability, the reviewers appreciated the focus of this work. The reviewers also found the experiments, both on real and synthetic datasets, extremely thorough and were impressed by the results. Finally, they also mentioned that the paper was clearly well-written.

The reviewers and AC note the following potential weaknesses:
(1) The primary concern, raised by all of the reviewers, is the lack of novelty;the proposed approach is a straightforward application of GAMs to model distillation, where black box output is the training data of the GAM, (2) The reviewers are also concern that the proposed approach is limited in scope to tabular datasets, and would not work for more interesting, complex domains like text or images, and (3) The reviewers are concerned that the interpretability of GAMS is assumed, without describing the limitations, for example, if there are correlated features, the shapes would affect each other in uninterpretable ways. Amongst other concerns, the reviewers were concerned about the formatting of the plots and tables in the paper, which made it difficult to read them, and the lack of a user study to verify the interpretability claims.

In response to these criticisms, the authors provided comments and a substantial revision to the papers, heavily restructuring the paper to fit extra experiments (comparison to other global explanation techniques, including a user study) and make the figures and tables readable. While the paper was much improved by these changes, and two of the reviewers increased their scores accordingly, concerns about the limited novelty and scope still remained.

Ultimately, the reviewers did not reach a conclusion, but the concerns of novelty and scope overwhelmed the clear benefits of the approach and the strong results. This paper was very close to getting accepted, and we strongly urge the authors to submit it to other premier ML conferences.